# Bayesian Phylodynamic Analysis Reveals the Dispersal Patterns of African Swine Fever Virus

**DOI:** 10.3390/v14050889

**Published:** 2022-04-25

**Authors:** Zhao-Ji Shen, Hong Jia, Chun-Di Xie, Jurmt Shagainar, Zheng Feng, Xiaodong Zhang, Kui Li, Rong Zhou

**Affiliations:** 1State Key Laboratory of Animal Nutrition, Key Laboratory of Animal Genetics Breeding and Reproduction, Ministry of Agriculture and Rural Affairs, Institute of Animal Science, Chinese Academy of Agricultural Sciences, Beijing 100193, China; xc428548@gmail.com (Z.-J.S.); jiahong80@126.com (H.J.); xxiechundi@163.com (C.-D.X.); jrmt1997@sina.com (J.S.); 2Guangdong Provincial Key Laboratory of Animal Molecular Design and Precise Breeding, College of Life Science and Engineering, Foshan University, Foshan 528231, China; greatfz@126.com; 3College of Animal Science and Technology, Anhui Agricultural University, Hefei 230036, China; xdzhang1983@ahau.edu.cn; 4Genome Analysis Laboratory of the Ministry of Agriculture and Rural Affairs, Agricultural Genomics Institute at Shenzhen, Chinese Academy of Agricultural Sciences, Shenzhen 518124, China

**Keywords:** ASFV, dispersal, evolution, phylogeographic patterns

## Abstract

The evolutionary and demographic history of African swine fever virus (ASFV) is potentially quite valuable for developing efficient and sustainable management strategies. In this study, we performed phylogenetic, phylodynamic, and phylogeographic analyses of worldwide ASFV based on complete ASFV genomes, B646L gene, and E183L gene sequences obtained from NCBI to understand the epidemiology of ASFV. Bayesian phylodynamic analysis and phylogenetic analysis showed highly similar results of group clustering between E183L and the complete genome. The evidence of migration and the demographic history of ASFV were also revealed by the Bayesian phylodynamic analysis. The evolutionary rate was estimated to be 1.14 × 10^−5^ substitution/site/year. The large out-migration from the viral population in South Africa played a crucial role in spreading the virus worldwide. Our study not only provides resources for the better utilization of genomic data but also reveals the comprehensive worldwide evolutionary history of ASFV with a broad sampling window across ~70 years. The characteristics of the virus spatiotemporal transmission are also elucidated, which could be of great importance for devising strategies to control the virus.

## 1. Introduction

The domestic pig, *Sus scrofa f. domestica*, is among the most common livestock widely raised by humans, which not only gives it enormous agricultural importance but also makes it a valuable biomedical research model. However, pig production has suffered large economic losses because of the outbreak of African swine fever (ASF).

ASF is a highly contagious disease manifesting clinical symptoms such as hemorrhagic fever, with high morbidity and mortality. It is contracted through infection by the African swine fever virus (ASFV). ASFV has a double-stranded DNA genome of 170–193 Kb, which contains from 150 to 167 open reading frames (ORFs). One major (p72) and four minor capsid proteins (M1249L, p17, p49, and H240R) make up the ASFV capsid structure, which is composed of 17,280 proteins [1]. The atomic structure of p72 determines the possible conformational epitopes, which may explain the difference between ASFV and other nucleocytoplasmic large DNA viruses (NCLDV) [1]. Among these genes, the B646L gene, which encodes the capsid protein p72, has often been used to classify the various strains into 24 different genotypes [2]. Another structural protein, p54, is a 25-KDa polypeptide encoded by the E183L gene. The p54 protein hijacks the microtubule motor complex to mediate viral transport through direct binding to the cytoplasmic motor light chain [3]. It also plays a vital function in virus morphogenesis. The formation of the viral precursor membrane can be hampered by reducing p54 protein synthesis [4]. Furthermore, the p54 protein is widely employed in the study of the characteristics of ASFV molecular evolution and can be used to detect ASFV [5,6,7,8].

ASF was first identified and reported in British East Africa (Kenya Colony) more than 100 years ago [9]. It has been reported that genotype I of ASFV was introduced to Portugal in 1957 [10]. This outbreak was effectively controlled, but a second outbreak in 1960 resulted in ASF becoming endemic in Portugal and Spain until 1995. With extensive controls, ASF was eradicated in Europe, except for Sardinia, where the disease has remained endemic since 1978 [11,12]. In 2007, an outbreak of ASFV genotype II in Georgia was likely triggered by livestock eating infected trash from ships near the port [13]. ASF was discovered in a region close to Armenia in August 2007 and in Azerbaijan and Russia in November 2007. Outbreaks were reported in several parts of the European Union in 2014, including Poland, Lithuania, Latvia, and Estonia [13,14,15,16,17]. The virus has spread worldwide, and more than 20 countries/territories have reported new or ongoing outbreaks through immediate notifications and follow-up reports, of which 9 in Europe, 12 in Asia, and 5 in Africa, according to the current ASF situation report (http://www.oie.int (accessed on 10 September 2021)). Most recently, since July 2021, ASF outbreaks have been reported in the Dominican Republic and Haiti [18], resurfacing in the Caribbean islands for the first time since the last occurrences of infection in Haiti in 1984.

The first ASF outbreak caused by ASFV genotype II in China was detected on 3 August 2018 [19] and it has now been reported across all the provinces of China, leading to large losses for the pig breeding industry. The viral genomes from the ASF outbreaks in Shenyang [19], Anhui [20], and Wuhan [21] were subsequently deposited into genome databases, and more than 80 ASFV genomes have been published worldwide until now. Indirect transmission mediated via humans is often considered to be one way that the virus is transmitted and introduced [22,23,24,25]. Due to the absence of an effective vaccine, improving biosecurity on farms, including by sealing off affected areas and culling infected animals, is currently the best and only control measure available.

The evolutionary and demographic history of ASFV remains poorly understood, even though such information is potentially quite valuable for developing efficient and sustainable management strategies. However, the route that ASFV followed to enter China also remains a mystery. In this study, we performed phylogenetic, phylogeographic, and phylodynamic analyses of worldwide ASFV based on complete ASFV genomes, B646L gene, and E183L gene sequences obtained from NCBI (https://www.ncbi.nlm.nih.gov/ (accessed on 16 January 2021)) to understand the epidemiology of ASFV and the dynamics of its transmission throughout the world.

## 2. Materials and Methods

### 2.1. Genome Sequence Alignment of ASFV

We obtained all 79 ASFV genomic sequences from the NCBI GenBank database, including ASFV genotypes I, II, III, IV, V, VIII, IX, X, XX, and XXII, on 16 January 2021. In addition, information that included the name, GenBank ID, p72 genotype, host, country, and collection time of each ASFV isolate was listed according to the NCBI and literature. The strain Liv13/33 was initially isolated in 1983 from a tick in Livingstone, Zambia, Africa [26], but the NCBI database describes the country of the isolate’s outbreak as France (Table 1). The cell lines of the adapted strain BA71V and ASFV strain Wuhan 2019-2, which is 100% consistent with ASFV Wuhan 2019-1, were not used in this analysis. Genomic sequence alignment was conducted with MAFFT (v7.450) using the default parameters.

### 2.2. Annotation of ASFV Genomes

Sixteen genomes without annotation files in the NCBI database were annotated using GATU software [27]. The open reading frames (ORFs) in each genome were predicted using Geneious (v2021.0.1) software (https://www.geneious.com/ (accessed on 18 February 2021)) and are listed in a table (Appendix A). The annotated GFF format file can be found in the Appendix A.

### 2.3. Detection of Homologous Recombination

According to the annotated file, gene sequences were obtained from 77 ASFV genomes with Geneious software and comparatively analyzed using Clustal Omega (v1.2.2) which created by Fabian Sievers and Desmond G. Higgins (Dublin, Ireland) [28,29]. We calculated the pairwise homoplasy index using the neighbor-net method [30]. In addition, the recombination events were detected using seven methods, namely, RDP, GENECONV, Chimaera, Bootscan, MaxChi, SiScan, and 3Seq [31].

### 2.4. Phylogenetic Analysis

The maximum likelihood method was used to conduct the phylogenetic analysis of 77 isolates based on the E183L gene sequences, B646L gene sequences, and complete genome sequences. The substitution model was selected based on the AIC scores obtained using ModelFinder [32]. The TVM+F+G4, TIM3e+G4, and GTR+F+G3 substitution models were applied to B646L, E183L, and the complete genome, respectively. An ultrafast bootstrap with 1000 replicates and the Shimodaira–Hasegawa approximate likelihood-ratio test (SH-aLRT) with 1000 replicates in IQ-TREE (v1.6.12) were used to evaluate the node support rate of the phylogenetic tree [33]. Additionally, these three datasets were recalculated using PhyML [34] to ensure the accuracy of the maximum likelihood tree.

### 2.5. Time Signal Detection and Evolutionary Mutation Rate

To determine the time scale of ASFV evolution, a date-randomized test was used to evaluate a molecular clock based on the complete genome. First, a date-randomized test (DRT) was performed to detect time signals in the data set [35].

The 15 permutations of the sampling dates were produced by the TipDatingBeast R package [36]. All date-randomized replicates and the initial file were generated using a Bayesian phylogenetic approach in BEAST 1.10.4 [37]. The Bayesian evaluation of Temporal Signal (BETS) is based on a Bayesian model comparison also used to judge whether there is a temporal signal. The generalized stepping-stone sampling (GSS) method in BEAST 1.10.4 was used to compare the heterochronous model and the isochronous model with a marginal likelihood estimation (MLE) [38,39]. The mean evolutionary rate of ASFV was estimated by LSD [40].

### 2.6. Population Dynamic

The best substitution model of the dataset was selected using ModelFinder. According to the Bayesian information criterion, four different model combinations, including different molecular clock models (uncorrelated relaxed clock and strict clock) and priori models of the tree (Bayesian skyline and constant size), were compared in BEAST 1.10.4 using path sampling and stepping-stone sampling, with the model for each combination running for an initial 10,000,000 states. In the next analysis, we used the uncorrelated relaxed clock with a Bayesian skyline plot to investigate the demographic history of ASFV. The MCMC was set at 10,000 steps over 2 × 10^8^ steps and run more than twice. The maximum clade credibility tree was calculated using TreeAnnotator v 1.8 (http://beast.bio.ed.ac.uk/TreeAnnotator/ (accessed on 25 May 2021)), and then a time-scaled the maximum clade credibility (MCC) tree was visualized by using FigTree v 1.4 software (http://tree.bio.ed.ac.uk/software/figtree/ (accessed on 25 May 2021)).

### 2.7. Genetic Differentiation among Different ASFV Populations

According to the geographical division of the world, we defined 11 populations of ASFV (Table 1), distrubuted in Central Africa (CAF), Central Europe (CE), East Africa (EAF), East Asia (EAS), East Europe (EE), South Africa (SAF), Southeast Asia (SEAS), Southern Europe (SE), West Africa (WAF), West Asia (WAS), and Western Europe (WE). The DnaSP v6.12 [41] was used to calculate pairwise *F*_ST_, which was the index used to measure the genetic differentiation among populations and to determine geneflow levels. Moderate differentiation (0.05–0.15), broad differentiation (0.15–0.25), great differentiation (>0.25), and infrequent gene flow (>0.33) were compared to explain how various ranges of the *F*_ST_ value affect the degrees of differentiation between populations.

### 2.8. Phylogeographic Analysis

To make educated guesses about the global spread of ASFV after the outbreak, the spatial propagation patterns were reconstructed in BEAST as a phylogeographic study. The latitude and longitude of ASFV regions in the dataset were obtained through Google Maps (https://www.google.com/maps (accessed on 30 May 2021)). With the Bayesian stochastic search variable selection (BSSVS) model in BEAST, we built a phylogeographic tree with discrete features based on the region’s latitude and longitude coordinates and the E183L gene sequence. We also estimated the number of expected region–state transitions (Markov jump counts) [42]. BSSVS was run three times independently to confirm the reliability of the analysis [43]. Then, after discarding a 10% burn-in, the results were entered into the SPREAD3 v 0.9.7 program [44] and visualized in Hyper-text Markup Language (HTML) format. To measure the existence of structures in the evolutionary diffusion of the virus, which is induced by the chosen discrete phenotype, the Ai statistic was measured using Bayesian Tip-Significance Testing (BaTS) version 1.0 [45].

## 3. Results

### 3.1. ASFV Genome Sequences

All 79 ASFV genomic sequences were obtained from the NCBI GenBank database on 16 January 2021. The length of the genome in the dataset ranges from 166,931 bp to 193,886 bp (Table 1) and is from 4 kinds of host and 23 countries. We identified 10 genotypes according to the difference in nucleotide sequences of the B646L gene, namely, I, II, III, IV, V, VIII, IX, X, XX, and XXII. The prevalent ASFV strains in EAS, SEAS, EE, WAS, WE, and CE were mainly genotype II, while the ASFV strains in SE, WAF, SAF, and EAF were mainly genotype I. Genotypes III, IV, V, VIII, IX, X, XX, and XXII were distributed in SAF, EAF, and CAF (Figure 1A).

Sixteen genomes without annotation files in the NCBI database were annotated using GATU software [27]. The open reading frames (ORFs) in each genome were predicted with Geneious (v2021.0.1) software and are listed in a table (Appendix A). The annotated GFF format file can be found in the Appendix A. Four isolates from ticks, nine isolates from pigs, two isolates from wild boars, and one isolate from a warthog were annotated. The 16 annotated isolates came from Malawi, South Africa, Namibia, Russia, Kenya, South Korea, and Poland. In total, 2191 ORFs were obtained in the 16 annotations, with an average of 137 ORFs in each isolate.

### 3.2. Phylogenetic Inference

With a high degree of support, where SH-aLRT was ≥75% and the bootstrap support value was ≥95%, the nodes are shown in the maximum-likelihood phylogenetic trees (ML trees) in Figure 1B–D. Of 25 p72 genotype I ASFV strains, 23 strains could be clustered into a clade based on E183L, B646L, and the complete genome. The isolates of p72 genotypes III, IV, I, XXII, VIII, V, and XX were mainly from ticks, wild boars, and warthogs in southern Africa and could be clustered together in the trees constructed using E183L and the complete genome sequences. Those multiple genotypes of ASFV may have been caused by its initial spread in South Africa and East Africa with a sylvatic cycle [46]. In addition, we calculated the correlation of the nucleotide sequence identities matrices based on the sequences of E183L, B646L, and the complete genome. The value of the correlation coefficient between E183L and the complete genome was 0.21230, while the value between B646L and the complete genome was 0.12886, which revealed that E183L and the complete genome resulted more consistently in group clustering compared with B646L.

### 3.3. Evolutionary Mutation Rate and Bayesian Phylodynamic Analysis

The Bayesian phylodynamic analysis was performed based on the sequence of E183L because of the more consistent result of group clustering between E183L and the complete genome in the phylogenetic analysis. Single gene sequences have been widely used to elucidate the system dynamic characteristics of viral epidemics and viruses’ evolution [43,47], and the E183L gene was used for phylogenetic analysis [48,49,50]. The results of the clustering are shown in the maximum clade credibility (MCC) tree according to the Bayesian phylodynamic analysis (Figure 2A) and were highly consistent with the results of the phylogenetic analysis (Figure 1B). To elucidate the evolutionary rate and population dynamic of ASFV, four different model combinations, including two molecular clock models (uncorrelated relaxed clock and strict clock) and two a priori models of the tree (Bayesian skyline and constant size) were compared, and then, the Bayesian skyline was selected and applied (Figure 2B), as it can precisely recover past bottlenecks in population size based on multi-locus data from a small number of individuals. Increasing the number of sampled individuals per locus could improve the estimates, but the effect was much more modest. The relative genetic diversity of worldwide ASFV is shown in the Bayesian skyline plot and illustrates the effective population size of these isolates. The population size of worldwide ASFV was relatively constant before 1975, but then it slowly decreased between 1975 and 1995 and sharply declined after 2000, which revealed that the relative genetic diversity of ASFV viruses decreased. The evolutionary rate of ASFV was estimated by LSD to be 1.14 × 10^−5^ substitution/site/year.

### 3.4. Migration of ASFV in the World

To explicate the migration and demographic history of ASFV around the world, 11 populations of ASFV were defined based on geographical division. The pairwise *F*_ST_ value was calculated among the 11 populations. There was great differentiation among the populations, except for the 19 pairwise regions with the value of *F*_ST_ < 0.33, which indicated that there were higher spatial genetic differentiation levels of ASFV in East Africa, Southern Europe, Center Africa, and West Africa populations compared to other regions (Figure 3C). This indicated no gene flow between p72 genotype I and genotype II.

The spatial propagation patterns (Figure 3A) revealed an epidemic of ASFV with an outbreak in SAF, which then spread widely into Southern Europe (SE) and Western Asia (WAS). ASF was transmitted throughout the European region (EE, CE, and WE) after 2007, and then the virus spread to the Asian region (EAS and SEAS). The SAF region experienced the largest out-migration of ASFV, and Europe (EE, CE, and WE) experienced the largest in-migration. To confirm the structure of the evolutionary diffusion of the ASFV, the Bayesian Tip-Significance Testing was performed; the *p*-value of the defined population was <0.01 or <0.05 and >0.01, except for CAF, WAF, and WE (Table 2). This result largely refers to the population structure defined by geographical location.

The regions with the largest out-migration of ASFV were SAF, EE, and CE. SAF, which was in the sylvatic cycle and has a larger number of ASFV hosts, such as warthogs and ticks, was, as expected, among those with the largest out-migration and diversity of viral genotypes. It was shown that the in-migration of ASFV was the largest in Europe (EE, CE, and WE), which may have been due to the ease of transportation between neighboring territories of the European Schengen countries (Figure 3B). The region of SEAS, which includes Vietnam, has only recently been affected by the pandemic and is the last region on the periphery of the continental landmass [51].

## 4. Discussion

Overall, we investigated the molecular epidemiology of worldwide ASFV based on published viral genomes. However, only 79 ASFV isolates were available from public databases in January 2021, the majority of which were strains of genotypes I and II. Moreover, only 10 genotype isolates with complete genomes could be downloaded from the public database, while the other 14 genotype isolates were reported only as single gene sequences or epidemic events, which made tracing the origin and evolution of ASFV complex. In addition, the absence of annotation files for published genomes also complicated the bioinformatic analysis. To better understand the dispersal patterns of ASFV, we not only downloaded all the available genomes from NCBI but also annotated 16 genomes that are provided in the Appendix A. Our phylogenetic analysis showed that the more consistent result of group clustering was between E183L and the complete genome and that the phylogenetic analysis with E183L and the complete genome was similar to the Bayesian phylodynamic analysis of E183L. Interestingly, LIV_5_40 (MN318203) was spilt with other genotype I isolates and is closer to multiple genotypes of ASFV in South Africa and East Africa. This may have been caused by the initial spread in South Africa and East Africa with the sylvatic cycle [46].

The evolutionary rate was compared with those of other family members of the nucleocytoplasmic large DNA viruses (NCDLV). ASFV evolutionary rate (1.14 × 10^−5^ substitution/site/year) was higher than those of Pithovirus sibericum (2.23 × 10^−6^ substitution/site/year) and Variola virus (6.5 × 10^−6^ substitution/site/year). However, it was reported that there is generally a negative association between the rates of evolutionary change and the genome size of microorganisms and DNA viruses [52,53]. Thus, it could be inferred that the actual evolutionary rate of ASFV should be lower than that estimated by LSD.

Furthermore, evidence of the migration and demographic history of worldwide ASFV was also lacking. It can be inferred that the EAF population experienced infrequent gene flow with the other 10 populations, with the *F*_ST_ value > 0.33. In addition, most of SE, CAF, and WAF also showed a high degree of differentiation compared to other regions. This is consistent with the results of the spatial diffusion of ASFV. It could be speculated that viruses with multiple genotypes from CAF and WAF have not spread to any part of the Eurasian region except SE (Portugal and Spain). This also confirmed the epidemiological history of the early outbreaks of ASFV in Europe. The higher genetic distances between the different regions also fit the evolutionary relationships represented on phylogenetic trees. Unexpectedly, these five regions with the same p72 genotype display an Fst value of 0 when compared. The strains that broke out in Eastern Asia showed a closer genetic distance to those in Africa and Southern Europe than to those of the other four regions.

The spatial propagation patterns were reconstructed in BEAST as a phylogeographic study. A migration pathway from SAF to WAS was observed, which coincided with the speculation that the ASFV outbreak in Georgia in 2007 was introduced from Africa [54]. The ASFV spread from the Caucasus to Russia, Belarus, Azerbaijan, Ukraine, etc., through wild boar populations caused large outbreaks among domestic pigs [55,56] and then reached Poland, Hungary, and Belgium. Unexpectedly, it could be inferred that the outbreak in EAS was the result of virus migration from CE, while the first outbreak of ASFV, ASFV-SY18 (MH766894), was reported in the northeast of China, much closer to Russia [19,57]. It could have been introduced by long-distance transmissions, such as through contaminated pork products [58], and by indirect transmission through humans [23,24,25].

The SAF population of ASFV identified with the largest out-migration had a crucial role in spreading the virus north to EAF and dispersing it to the SE and WAF regions, crossing the geographical barrier of the African mainland. In addition, the analysis of the size of the out-migration of ASFV by a Markov jump estimate also showed that the EE and CE populations underwent a larger out-migration than an in-migration. This is also consistent with the reported epidemiological history of the type II strains since the Georgia outbreak in 2007. ASFV has spread from the Caucasus to Russia, Belarus, Azerbaijan, Ukraine, etc., through wild boar populations and has caused large outbreaks among domestic pigs. The mixed infection of domestic and wild boar populations has been going on for many years, being a central source of the spread of the current pandemic to the rest of Europe and Asia [55,56].

However, despite the improvement reported in this study using the E183L gene as a target for the presented analysis, these results should be taken with caution, considering the still low correlation observed between the topologies displayed by the phylogenetic trees developed using full-length sequences and E183L. This low correlation can be clearly explained by the complex evolutionary process observed for different genes of ASFV [59,60,61]. In this context, we consider that further analysis should be performed using full-length sequences.

## 5. Conclusions

Our study provides a comprehensive evolutionary history of worldwide ASFV with a broad sampling window across ~70 years and elucidates the characteristics of the virus’ spatiotemporal transmission, which could be a crucial reference for the management of strategies to control the virus.

## Figures and Tables

**Figure 1 viruses-14-00889-f001:**
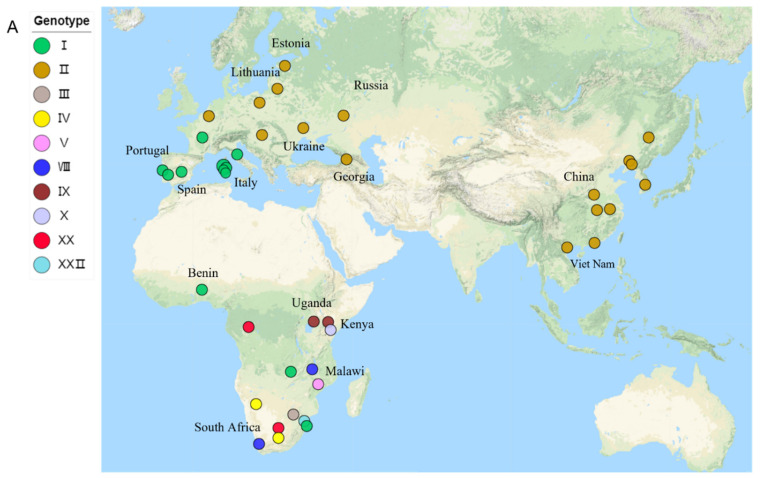
Phylogenetic relationships of the African swine fever virus isolates: (**A**) geographical distribution of ASFV in this study; (**B**) maximum-likelihood tree of 77 ASFV isolates based on complete genome sequences; (**C**) maximum-likelihood tree of 77 ASFV isolates based on E183L sequences; (**D**) maximum-likelihood tree of 77 ASFV isolates based on B646L. Different colors represent various p72 genotypes.

**Figure 2 viruses-14-00889-f002:**
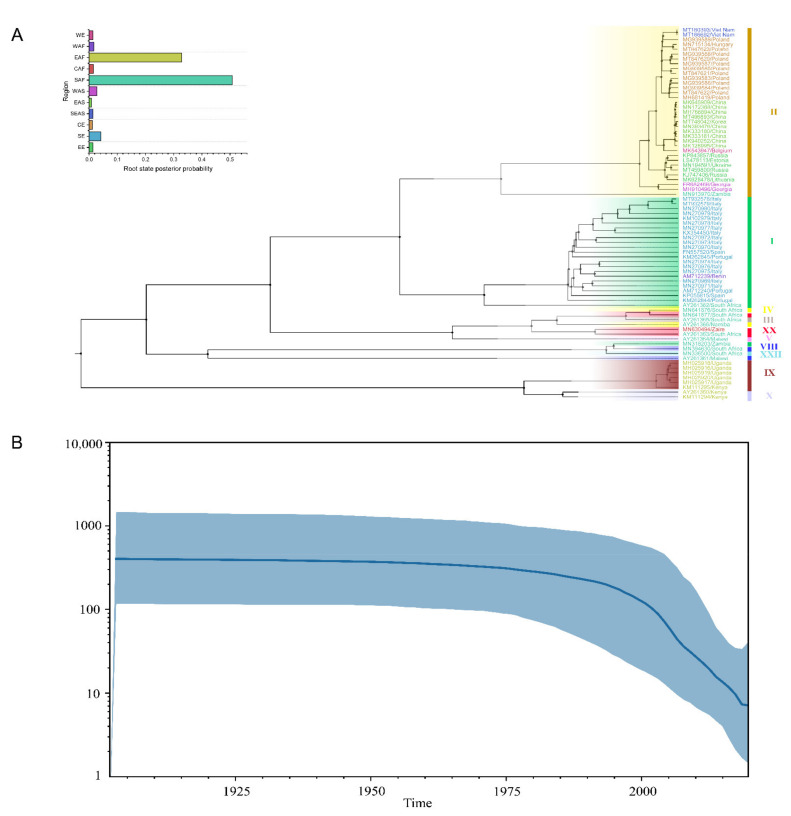
Bayesian phylodynamic analysis of ASFV: (**A**) maximum clade credibility tree of the E183L gene of ASFV; different colors indicate the virus location—the histogram represents the posterior probability of the root state in the MCC tree; (**B**) Bayesian skyline plot showing the population size in time for worldwide ASFV. The y-axis represents the effective population size (Ne) and the virus generation time (τ). The x-axis reports the time. The dark blue line shows the median estimate of the population size, and the light blue shading shows the 95% credibility interval.

**Figure 3 viruses-14-00889-f003:**
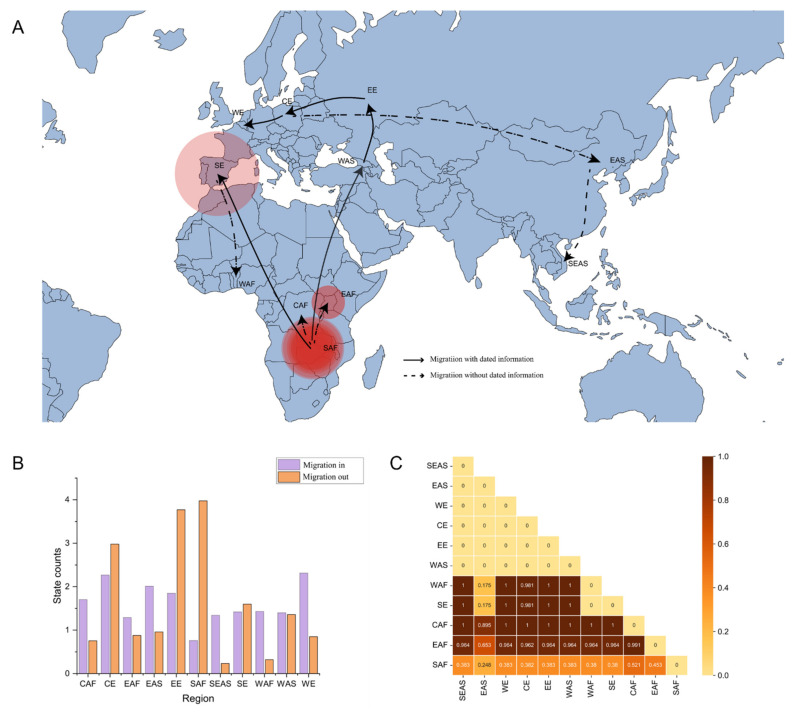
Spatial diffusion of ASFV: (**A**) spatial diffusion pathway of ASFV. The dark lines with arrows represent the transmission pathway of ASFV. The solid line indicates the known ASFV migration pathway, and the dashed line indicates the unproven ASFV migration pathway. (**B**) The purple color represents ASFV migration into the region, and the orange color represents ASFV migration out of the region. (**C**) Pairwise *F*_ST_ of ASFV populations.

**Table 1 viruses-14-00889-t001:** ASFV isolates used in this study.

GenBank ID	Isolate	Host	Country	Population	Year	p72	Length (bp)
						Genotype	
MN318203	LIV_5_40	Tick	Zambia	SAF	1983	I	183,291
MN630494	Zaire	Pig	Zaire	CAF	1977	XX	185,338
MT166692	ASFV_Hanoi_2019	Pig	Viet Nam	SEAS	2019	II	166,931
MT180393	ASFV_NgheAn_2019	Pig	Viet Nam	SEAS	2019	II	186,498
MN194591	ASFV/Kyiv/2016/131	Pig	Ukraine	EE	2016	II	191,911
MH025919	N10	Pig	Uganda	EAF	2015	IX	188,611
MH025918	R25	Pig	Uganda	EAF	2015	IX	188,630
MH025920	R35	Pig	Uganda	EAF	2015	IX	188,629
MH025917	R7	Pig	Uganda	EAF	2015	IX	188,628
MH025916	R8	Pig	Uganda	EAF	2015	IX	188,627
KP055815	BA71	Pig	Spain	SE	1971	I	180,365
FN557520	E75	Pig	Spain	SE	1975	I	181,187
ASU18466	BA71V	Vero cells	Spain	SE	1971	I	170,101
MT748042	ASFV/Korea/pig/PaJu1/2019	Pig	South Korea	EAS	2019	II	190,597
AY261362	Mkuzi 1979	Tick	South Africa	SAF	1979	I	192,714
MN394630	SPEC_57	Tick	South Africa	SAF	1985	VIII	186,119
AY261363	Pretorisuskop/96/4	Tick	South Africa	SAF	1996	XX	190,324
MN641876	RSA_W1_1999	Warthog	South Africa	SAF	1999	IV	185,293
MN641877	RSA_2_2004	Wild boar	South Africa	SAF	2004	XX	188,502
MN336500	RSA_2_2008	Tick	South Africa	SAF	2008	XXII	187,866
AY261365	Warmbaths	Tick	South Africa	SAF	1987	III	190,773
KJ747406	Kashino 04/13	Wild boar	Russia	EE	2013	II	189,387
KP843857	Odintsovo_02/14	Pig	Russia	EE	2014	II	189,333
MT459800	ASFV/Kabardino-Balkaria 19/WB-964	Wild boar	Russia	EE	2019	II	189,252
KM262844	L60	Pig	Portugal	SE	1960	I	182,362
KM262845	NHV	Pig	Portugal	SE	1968	I	172,051
AM712240	OURT 88/3	Tick	Portugal	SE	1988	I	171,719
MH681419	ASFV/POL/2015/Podlaskie	Wild boar	Poland	CE	2015	II	189,394
MG939584	Pol16_20538_o9	Pig	Poland	CE	2016	II	189,399
MG939585	Pol16_20540_o10	Pig	Poland	CE	2016	II	189,405
MG939586	Pol16_29413_o23	Pig	Poland	CE	2016	II	189,393
MG939583	Pol16_20186_o7	Pig	Poland	CE	2016	II	189,401
MG939587	Pol17_03029_C201	Pig	Poland	CE	2017	II	189,405
MG939588	Pol17_04461_C210	Pig	Poland	CE	2017	II	189,401
MG939589	Pol17_05838_C220	Pig	Poland	CE	2017	II	189,393
MT847620	Pol17_55892_C754	Pig	Poland	CE	2017	II	189,414
MT847621	Pol18_28298_O111	Pig	Poland	CE	2018	II	189,409
MT847623	Pol19_53050_C1959/19	Pig	Poland	CE	2019	II	189,356
MT847622	Pol17_31177_O81	Pig	Poland	CE	2017	II	189,422
AY261366	Warthog	Warthog	Namibia	SAF	1980	IV	186,528
AY261364	Tengani 62	Pig	Malawi	SAF	1962	V	185,689
AY261361	Malawi Lil-20/1 (1983)	Tick	Malawi	SAF	1983	VIII	187,612
MK628478	ASFV/LT14/1490	Wild boar	Lithuania	EE	2014	II	189,399
AY261360	Kenya 1950	Pig	Kenya	EAF	1950	X	193,886
KM111294	Ken05/Tk1	Tick	Kenya	EAF	2005	X	191,058
KM111295	Ken06.Bus	Pig	Kenya	EAF	2006	IX	184,368
MN270969	56/Ca/1978	Pig	Italy	SE	1978	I	183,636
MN270970	57/Ca/1979	Pig	Italy	SE	1979	I	183,639
MN270971	139/Nu/1981	Pig	Italy	SE	1981	I	183,645
MN270972	140/Or/1985	Pig	Italy	SE	1985	I	183,723
MN270973	85/Ca/1985	Pig	Italy	SE	1985	I	181,816
MN270974	141/Nu/1990	Pig	Italy	SE	1990	I	183,720
MN270975	142/Nu/1995	Pig	Italy	SE	1995	I	183,724
MN270976	60/Nu/1997	Pig	Italy	SE	1997	I	181,651
MN270977	26/Ss/2004	Pig	Italy	SE	2004	I	184,581
MN270978	72407/Ss/2005	Pig	Italy	SE	2005	I	181,699
KX354450	47/Ss/2008	Pig	Italy	SE	2008	I	184,638
KM102979	26544/OG10	Pig	Italy	SE	2010	I	182,906
MN270979	97/Ot/2012	Pig	Italy	SE	2012	I	184,206
MN270980	22653/Ca/2014	Pig	Italy	SE	2014	I	181,869
MT932578	103917/18	Pig	Italy	SE	2018	I	181,759
MT932579	55234/18	Pig	Italy	SE	2018	I	181,761
MN715134	ASFV_HU_2018	Wild boar	Hungary	CE	2018	II	190,601
FR682468	Georgia 2007/1	Pig	Georgia	WAS	2007	II	189,344
MH910496	Georgia 2008/2	Pig	Georgia	WAS	2008	II	189,315
MN913970	Liv13/33	Tick	Zambia	SAF	1983	I	188,277
LS478113	Estonia 2014	Wild boar	Estonia	EE	2014	II	182,446
MK645909	ASFV-wbBS01	Wild boar	China	EAS	2018	II	189,394
MK128995	China/2018/AnhuiXCGQ	Pig	China	EAS	2018	II	189,393
MK333180	Pig/HLJ/2018	Pig	China	EAS	2018	II	189,404
MH766894	ASFV-SY18	Pig	China	EAS	2018	II	189,354
MK333181	DB/LN/2018	Pig	China	EAS	2018	II	189,404
MN172368	ASFV/pig/China/CAS19-01/2019	Pig	China	EAS	2019	II	189,405
MN393476	ASFV Wuhan 2019-1	Pig	China	EAS	2019	II	190,576
MN393477	ASFV Wuhan 2019-2	Pig	China	EAS	2019	II	190,576
MK940252	CN/2019/InnerMongolia-AES01	Wild boar	China	EAS	2019	II	189,403
MT496893	GZ201801	Pig	China	EAS	2018	II	189,393
AM712239	Benin 97/1	Pig	Benin	WAF	1997	I	182,284
MK543947	Belgium/Etalle/wb/2018	Wild boar	Belgium	WE	2018	II	190,202

The 11 populations of ASFV were defined by geographical division, namely, Central Africa (CAF), Central Europe (CE), East Africa (EAF), East Asia (EAS), East Europe (EE), South Africa (SAF), Southeast Asia (SEAS), Southern Europe (SE), West Africa (WAF), West Asia (WAS), and Western Europe (WE).

**Table 2 viruses-14-00889-t002:** Phylogeny–trait association test of the geographic structure of ASFV by Bayesian Tip-Significance Testing.

Statistic	Observed Mean (95% CI)	Null Mean (95% CI)	*p*-Value
AI	0.94 (0.55–1.32)	6.86 (6.21–7.40)	0
PS	12.48 (11.00–14.00)	49.92 (47.24–52.10)	0
MC (EAF)	8.00 (8.00–8.00)	1.21 (1.01–1.96)	<0.01
MC (SE)	10.89 (6.00–18.00)	2.06 (1.60–2.73)	<0.01
MC (SAF)	3.66 (3.00–4.00)	1.38 (1.07–1.97)	<0.01
MC (CAF)	1.00 (1.00–1.00)	1.00 (1.00–1.00)	>0.05
MC (WAF)	1.00 (1.00–1.00)	1.00 (1.00–1.00)	>0.05
MC (WAS)	1.70 (1.00–2.00)	1.00 (1.00–1.01)	<0.01
MC (EE)	2.55 (1.00–6.00)	1.13 (1.00–1.57)	<0.05 & >0.01
MC (CE)	6.34 (3.00–12.00)	1.56 (1.18–2.16)	<0.01
MC (WE)	1.00 (1.00–1.00)	1.00 (1.00–1.03)	>0.05
MC (EAS)	7.22 (3.00–10.00)	1.31 (1.04–2.00)	<0.01
MC (SEAS)	1.98 (2.00–2.00)	1.00 (1.00–1.00)	<0.01

## Data Availability

The annotated GFF format file of 16 isolates can be found in the Appendix A.

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
