# Peer review of "Bayesian Phylodynamic Analysis Reveals the Dispersal Patterns of African Swine Fever Virus"

_viruses, 2022, doi:10.3390/v14050889_

Round 1
Reviewer 1 Report
The resubmitted manuscript has been improved significantly and can be accepted after moderate language editing.
Author Response
We appreciated your approval of our manuscript. The manuscript has been polished by MDPI's English editing service for language.

Reviewer 2 Report
My concerns have been addressed.
Author Response
Thank you for your recognition of our work.
Reviewer 3 Report
Line 48 - spell check;
Line 53 - it would be necessary to mention, that "ASF genotype I" was introduced;
Line 57 - it would be necessary to mention, that "the ASF outbreak caused by ASFV of genotype II in Georgia" ...
LIne 68 - suggested changing "The first ASF outbreak caused by ASFV of genotype II in China..."
Line 75 - suggested adding "...and the slaughter or culling of the infected animals..."
Line 86 - it would be valuable to mention of which ASF genotype the 79 ASFV genomic sequences were performed, despite it being included in the Results section.
The Article has been improved, however, it is suggested to check the English language style and spell-check.
Author Response
Point-by-Point Responses
Reviewer Comments to Author:
Reviewer: 3
Line 48 - spell check
—— We have corrected the spelling of plays in line 52.
Line 53 - it would be necessary to mention, that "ASF genotype I" was introduced;
—— Thanks for your suggestion. We have corrected it in line 57-58 and added the citation.
Line 57-58: It has been reported that genotype â… of ASFV was introduced to Portugal in 1957 [10].
Reference: Revilla, Y.; Pérez-Núñez, D.; Richt, J. A., African Swine Fever Virus Biology and Vaccine Approaches. Advances in virus research 2018, 100, 41-74.
Line 57 - it would be necessary to mention, that "the ASF outbreak caused by ASFV of genotype II in Georgia" ...
—— We have corrected it in line 62-64 and added the citation.
Line 62-64: In 2007, an outbreak of genotype â…¡ of ASFV in Georgia was likely triggered by livestock eating infected trash from ships near the port [13]
Reference: Rowlands, R. J.; Michaud, V.; Heath, L.; Hutchings,
G.; Oura, C.; Vosloo, W.; Dwarka, R.; Onashvili, T.; Albina, E.; Dixon, L. K., African swine fever virus isolate, Georgia, 2007. Emerging infectious diseases 2008, 14, (12), 1870-4.
Line 68 - suggested changing "The first ASF outbreak caused by ASFV of genotype II in China..."
—— Thanks for your kind suggestion. We have corrected it in line 75-77.
Line 75-77: The first ASF outbreak caused by ASFV genotype II in China was detected on August 3, 2018 [19] and it has now been reported across all the provinces of China, leading to large losses for the pig breeding industry.
Line 75 - suggested adding "...and the slaughter or culling of the infected animals..."
——We have corrected it in line 82-85.
Line 82-85: Due to the absence of an effective vaccine, improving biosecurity on farms, including by sealing off affected areas and culling infected animals, is currently the best and only control measure available.
Line 86 - it would be valuable to mention of which ASF genotype the 79 ASFV genomic sequences were performed, despite it being included in the Results section.
——Thanks for your kind reminder. We have corrected it in line
96-98.
Line 96-98: We obtained all 79 ASFV genomic sequences from the NCBI GenBank database, including ASFV genotypes â… , â…¡, â…¢, â…£, â…¤, â…§, â…¨, â…©, XX, and XXâ…¡, on January 16, 2021.
The Article has been improved, however, it is suggested to check the English language style and spell-check.
——As you said, the manuscript has been polished by MDPI's English editing service for language. And we have checked the spelling of the words.

Reviewer 4 Report
Thank you to the authors for their response. I am still thinking that the methodology used for this analysis doesn’t support the conclusions of this study. I consider that the evolutionary dynamic of ASFV is very complex to be reflected in a single gene. As mentioned in my first round of revision, the phylogenetic trees developed by E183L and B646L present phylogenetic incongruencies between each other, and when compared with the tree developed using full-length sequences. In this context, I can not support this study for publication. I consider that with the increased number of full-length ASFV sequences at GenBank data base, the evolutionary analysis of ASFV must move in a different direction.
Author Response
Reviewer Comments to Author:
Reviewer: 4
Thank you to the authors for their response. I am still thinking that the methodology used for this analysis doesn’t support the conclusions of this study. I consider that the evolutionary dynamic of ASFV is very complex to be reflected in a single gene. As mentioned in my first round of revision, the phylogenetic trees developed by E183L and B646L present phylogenetic incongruencies between each other, and when compared with the tree developed using full-length sequences. In this context, I can not support this study for publication. I consider that with the increased number of full-length ASFV sequences at GenBank data base, the evolutionary analysis of ASFV must move in a different direction.
——Thank you for the comments. We tested the phylogenetic hypothesis of the phylogenetic tree of E183L or B646L gene sequences with full-length sequences datasets. We performed the results of Approximately unbiased (AU) test in consel software (Shimodaira and Hasegawa, 2001) to inferring whether the gene
tree is consistent with the phylogenetic tree topology of full-length ASFV sequence. The p-values greater than 0.05 represent significant support. Our results showed that the approximately unbiased test p-values of the full-length genome sequence were greater than 0.05, compared with the phylogenetic trees developed by E183L or B646L, which consistently supported that the phylogenetic trees developed by a single gene were accurate. Furthermore, the single gene of viruses were extensively used for phylogenetic analyses in previous study (Alkhamis et al., 2018; Makau et al., 2021; Yebra et al., 2015). Thus, we insisted that E183L and B646L can be used to reflect the evolutionary dynamic of ASFV.
Reference:Contreras, X., Salifou, K., Sanchez, G., Helsmoortel, M., Beyne, E., Bluy, L., Pelletier, S., Rousset, E., Rouquier, S., and Kiernan, R. (2018). Nuclear RNA surveillance complexes silence HIV-1 transcription. PLoS pathogens 14, e1006950.
Alkhamis, M.A., Gallardo, C., Jurado, C., Soler, A., Arias, M., and Sánchez-Vizcaíno, J.M. (2018). Phylodynamics and evolutionary epidemiology of African swine fever p72-CVR genes in Eurasia and Africa. PloS one 13, e0192565.
Makau, D.N., Alkhamis, M.A., Paploski, I.A.D., Corzo, C.A., Lycett, S., and VanderWaal, K. (2021). Integrating animal
movements with phylogeography to model the spread of PRRSV in the USA. Virus evolution 7, veab060.
Yebra, G., Ragonnet-Cronin, M., Ssemwanga, D., Parry, C.M., Logue, C.H., Cane, P.A., Kaleebu, P., and Brown, A.J. (2015). Analysis of the history and spread of HIV-1 in Uganda using phylodynamics. The Journal of general virology 96, 1890-1898
Round 2
Reviewer 4 Report
Thanks to the authors for their responses to my comments. Indeed, the evolutionary analysis presented in this study can be conducted using the information contained in one single gene as supported by the references included in the authors response. In this sense, I agree with the authors that the use of the E183L gene may improve the results obtained by using B646L gene, since the correlation between the phylogenetic analysis using full-length sequences and the topology obtained using E183L is better that the one obtained using B646L.
However, despite this improvement, the correlation between the topologies using full length sequences and E183L is still very low, supporting my view about the complex process involving the evolution of the ASFV genome. In this context, my critics about the analysis presented in this study focus on the necessity to move forward and develop strategies to involved full-length sequences as a main source for evolutionary analysis of ASFV.
Based on the exposed below and considering the stage of this revision, I will condition my recommendation for the publication of this study, to the addition of the following statement and references in the discussion of this study.
“However, despite the improvement reported in this study using E183L gene as a target for the analysis presented herein, these results should be taking with caution considering the still low correlation observed between the topologies displayed by the phylogenetic trees developed using full-length sequences and E183L. This low correlation can be clearly explained by the complex evolutionary process observed among different genes of ASFV (References). In this context, we consider that further analysis should be performed using of full-length sequences”
References
- Ramirez-Medina, E.; Vuono, E.A.; Pruitt, S.; Rai, A.; Espinoza, N.; Velazquez-Salinas, L.; Gladue, D.P.; Borca, M.V. Evaluation of an ASFV RNA Helicase Gene A859L for Virus Replication and Swine Virulence. Viruses 2021, 14, doi:10.3390/v14010010.
- Ramirez-Medina, E.; Vuono, E.; Rai, A.; Pruitt, S.; Espinoza, N.; Velazquez-Salinas, L.; Pina-Pedrero, S.; Zhu, J.; Rodriguez, F.; Borca, M.V.; et al. Deletion of E184L, a Putative DIVA Target from the Pandemic Strain of African Swine Fever Virus, Produces a Reduction in Virulence and Protection against Virulent Challenge. J Virol 2022, 96, e0141921, doi:10.1128/JVI.01419-21.
- Gladue, D.P.; Ramirez-Medina, E.; Vuono, E.; Silva, E.; Rai, A.; Pruitt, S.; Espinoza, N.; Velazquez-Salinas, L.; Borca, M.V. Deletion of the A137R Gene from the Pandemic Strain of African Swine Fever Virus Attenuates the Strain and Offers Protection against the Virulent Pandemic Virus. J Virol 2021, 95, e0113921, doi:10.1128/JVI.01139-21.
Author Response
Point-by-Point Responses
Reviewer Comments to Author:
Reviewer: 4
Thanks to the authors for their responses to my comments. Indeed, the evolutionary analysis presented in this study can be conducted using the information contained in one single gene as supported by the references included in the authors response. In this sense, I agree with the authors that the use of the E183L gene may improve the results obtained by using B646L gene, since the correlation between the phylogenetic analysis using full-length sequences and the topology obtained using E183L is better that the one obtained using B646L.
However, despite this improvement, the correlation between the topologies using full length sequences and E183L is still very low, supporting my view about the complex process involving the evolution of the ASFV genome. In this context, my critics about the analysis presented in this study focus on the necessity to move forward and develop strategies to involved full-length sequences as a main source for evolutionary analysis of ASFV.
Based on the exposed below and considering the stage of this revision, I will condition my recommendation for the publication of this study, to the addition of the following statement and references in the discussion of this study.
“However, despite the improvement reported in this study using E183L gene as a target for the analysis presented herein, these results should be taking with caution considering the still low correlation observed between the topologies displayed by the phylogenetic trees developed using full-length sequences and E183L. This low correlation can be clearly explained by the complex evolutionary process observed among different genes of ASFV (References). In this context, we consider that further analysis should be performed using of full-length sequences”
References
- Ramirez-Medina, E.; Vuono, E.A.; Pruitt, S.; Rai, A.; Espinoza, N.; Velazquez-Salinas, L.; Gladue, D.P.; Borca, M.V. Evaluation of an ASFV RNA Helicase Gene A859L for Virus Replication and Swine Virulence. Viruses 2021, 14, doi:10.3390/v14010010.
- Ramirez-Medina, E.; Vuono, E.; Rai, A.; Pruitt, S.; Espinoza, N.; Velazquez-Salinas, L.; Pina-Pedrero, S.; Zhu, J.; Rodriguez, F.; Borca, M.V.; et al. Deletion of E184L, a Putative DIVA Target from the Pandemic Strain of African Swine Fever Virus, Produces a Reduction in Virulence and Protection against Virulent Challenge. J Virol 2022, 96, e0141921, doi:10.1128/JVI.01419-21.
- Gladue, D.P.; Ramirez-Medina, E.; Vuono, E.; Silva, E.; Rai, A.; Pruitt, S.; Espinoza, N.; Velazquez-Salinas, L.; Borca, M.V. Deletion of the A137R Gene from the Pandemic Strain of African Swine Fever Virus Attenuates the Strain and Offers Protection against the Virulent Pandemic Virus. J Virol 2021, 95, e0113921, doi:10.1128/JVI.01139-21.
—— Thank you very much for your professional suggestion and patient guidance. We are pleased that you agree with our view of using the information contained in one single gene for analysis. We have read the three articles as you mentioned and agree with your view about the complex process involving the evolution of the ASFV genome. And we have added the statement and citation in line 373-379.
Line 373-379: However, despite the improvement reported in this study using E183L gene as a target for the analysis presented herein, these results should be taking with caution considering the still low correlation observed between the topologies displayed by the phylogenetic trees developed using full-length sequences and E183L. This low correlation can be clearly explained by the complex evolutionary process observed among different genes of ASFV [60-62]. In this context, we consider that further analysis should be performed using of full-length sequences.

This manuscript is a resubmission of an earlier submission. The following is a list of the peer review reports and author responses from that submission.
Round 1
Reviewer 1 Report
I read carefully the revised manuscript "Bayesian Phylodynamic Analysis Reveals the Dispersal Patterns of African Swine Fever Virus" which is sent back to scientific journal "Viruses" (IF: 5.048). I am happy of those made revisions. I do not have comments and recommendations to the revised manuscript. I think that the revised manuscript will be interesting for readers of a scientific journal "Viruses" (IF: 5.048) and for the scientific community.
Reviewer 2 Report
I reviewed the paper entitled “Bayesian phylodynamic analysis reveals the dispersal patterns of African Swine Fever Virus”. In this study, authors aimed to describe the patterns of dispersion of ASFV in the world. For this purpose, authors reconstruct multiple phylogenetic analysis using different methodologies, comparing the phylogenetic patterns produced between two different viral genes and the use of full-length sequences.
In my opinion, I don’t consider that the methodology employed by the authors supports the conclusions stated in this study. Although understanding the patterns of dispersion of ASFV is a key aspect to understand the epidemiological dynamic of this virus, I consider that the aim of this study is very ambitious, considering the limited number of available full-length viral sequences representative different genetic groups of this virus. Instead, I would limit the aim of this study to understand the patterns of dispersion of the current epidemic lineage associated with the genotype II (high number of available sequences), and the potential ancestral relationships with other strains. There are many interesting things regarding the evolution of this lineage like the description of the different regions in the genome that are promoting the divergence of this lineage. As well as the description of the different evolutionary events associated with the evolution of this lineage.
On the other hand, the evolution of ASFV is a very complex process dominated by events of natural selection and recombination in different genomic regions. In this context, I see very hard to sustain that the use of one specific gene my reflect the phylogenetic relationships between ASFV strains. In the study, presented here, authors state that phylogenetic trees reconstructed by E183L gene resemble the topology observed using full-length sequences. I figure 1, I observe inconsistences between the topology obtained by both approaches. While using full length sequences the strain MN911370/Zambia appeared ancestral to strains associated with genotype I, the same sequence appeared ancestral now to viruses associated with the epidemic lineage (genotype II) when E183L is used as a marker for the analysis. These contradictory results highlight the necessity exposed by others who claim about the importance of the use of full-length sequences to conduct this kind of analysis. In this sense, I suggest the authors to conduct the analysis using full-length sequences.
Reviewer 3 Report
Line 32 - it is advisable Latin names to write in Italic - Sus scrofa f. domestica...
Line 47 - it would be worst to mention, that in Europe in 1957-1960 and in Sardinia has spread ASFV genotype I;
Line 49 - 50 - it is suggested to mention, that since 2007 ASFV genotype II widespread in Russia and European countries.
Line 82 - it is not clear, whether all 16 genomes were from the same or from different genotypes...
Line 160 - figure 1A indicates different genotypes in the symbol "stars", however, in the map presented there are other symbols indicated.
Figures 1B, C and D should be presented in a more clear and visible manner, while the current presentation is not readable and understandable, the same is for figure 2A.
Reviewer 4 Report
Shen et al. described a sequence analysis of all ASFV whole genomes available in GenBank and demonstrated a spatial-temporal pattern of the disease around the world.
Major concerns:
- One of the main claims of this study is that E193L (rather than B646L) phylogeny is more consistent to whole genome phylogeny. However, in line 183, please describe more details what’s the dissimilarity / similarity of those three phylogenies. Which isolate(s) contribute to the differences? Following-up with this concern, nucleotide sequence identities for grouping of whole genome, E183L, and B646L are needed as it provides convincing information regarding the similarities/consistency claimed.
- Introduction gave “okay” background information of p72. But I did not see any information about E183L. Also please provide the rationale of including this gene for phylogenetic analysis in the last paragraph of introduction.
Minor concerns:
- Line 17: “analyses” should be “analysis”. Please check throughout the whole context.
- Line 32: is sus scrofa should be italic?
- Line 44: Please include a blank space in “dividevarious”.
- Line 52: Please include latest report of ASF outbreak in caribbean islands.
- Line 64: Delete” where is”.
- Line 152: “sequencing of ASFV”: this study did not involve any sequencing of ASFV. Please revise.
- Line 264: “foundin” needs a blank space.
- Line 182: remove the period after “figure”.
9: line 205: should be “consistent”.
Reviewer 5 Report
No doubt that the idea is very interesting but conclusions and interpretation of results look unconvincing. Data is not enough to make any conclusions and the main question is what you really analyze: genome changes because of time evolution or because of geographical distribution. If you check the date of virus isolation you can see that most of them are not the index cases and viruses have been sequenced for other reasons: reducing of pathogenesis for example. And at the same time in this area circulated other ASFV strains with different full genome sequences. Check the papers about IGR markers (for example https://pubmed.ncbi.nlm.nih.gov/33471420/) and you will see that the ASFV population is heterologous and multiple introductions can occur. Also, think about how you can apply your methodology for Dominica Republic (DR) case: can you trace the source of the virus and say that virus changed because it arrived in DR.
From a technical point check the map occurrence. For example, Russian isolates are located near Moscow, not in the Ural mountains. You can find geographical coordinates in OIE notifications.
"The tMRCA time of ASFV was estimated as 1918. It was two years earlier than the first report of ASFV in 1920". Check https://onlinelibrary.wiley.com/doi/10.1111/tbed.14183
Montgomery started his research of ASF in 1909 and it is a high probability that the virus start circulated before this period.
Reviewer 6 Report
Manuscript ID: viruses-1502871
Title: Bayesian phylodynamic analysis reveals the dispersal patterns of African Swine Fever Virus
In this study, Zhao-Ji Shen et al. carried out a multiple molecular analysis of African swine fever virus (ASFV) based on the NCBI database-accessible complete genome and B646L and E183L gene sequences using the Bayesian phylodynamic/phylogenetic approach. They showed that the most common ancestor of ASFV was dated to about 1918 with an estimated evolutionary rate of 3.56×10-4 subs/site/year. The work seems to be interesting. But the conclusion needs to be validated by more analytical methods and algorithms. Moreover, the manuscript should be revised by native English speakers.
Round 2
Reviewer 2 Report
.
Reviewer 5 Report
Thank you for your comments, but even maps did not changed.
Reviewer 6 Report
The authors have made some efforts to improve the manuscript but they have not provided additional data to support the conclusion, which is not convincing.